

# The impacts of bovine milk, soy beverage, or almond beverage on the growing rat microbiome

Julie Cakebread[1,2], Olivia A.M. Wallace[1], Harold Henderson[1], Ruy Jauregui[3], Wayne Young[2] and Alison Hodgkinson[1]

[1] Food and Biobased Products, AgResearch Ltd., Hamilton, New Zealand
[2] Smart Foods Innovation Centre of Excellence, AgResearch Ltd., Palmerston North, New Zealand
[3] Digital Agriculture Innovation Centre of Excellence, AgResearch Ltd., Palmerston North, New Zealand

Corresponding author
Julie Cakebread,
Julie.cakebread@agresearch.co.nz

## ABSTRACT

**Background**. Milk, the first food of mammals, helps to establish a baseline gut microbiota. In humans, milk and milk products are consumed beyond infancy, providing comprehensive nutritional value. Non-dairy beverages, produced from plant, are increasingly popular as alternatives to dairy milk. The nutritive value of some plant-based products continues to be debated, whilst investigations into impacts on the microbiome are rare. The aim of this study was to compare the impact of bovine milk, soy and almond beverages on the rat gut microbiome. We previously showed soy and milk supplemented rats had similar bone density whereas the almond supplemented group had compromised bone health. There is an established link between bone health and the microbiota, leading us to hypothesise that the microbiota of groups supplemented with soy and milk would be somewhat similar, whilst almond supplementation would be different.

**Methods**. Three-week-old male Sprague Dawley rats were randomly assigned to five groups ($n = 10$/group) and fed *ad libitum* for four weeks. Two control groups were fed either standard diet (AIN-93G food) or AIN-93G amino acids (AA, containing amino acids equivalent to casein but with no intact protein) and with water provided *ad libitum*. Three treatment groups were fed AIN-93G AA and supplemented with either bovine ultra-heat treatment (UHT) milk or soy or almond UHT beverages as their sole liquid source. At trial end, DNA was extracted from caecum contents, and microbial abundance and diversity assessed using high throughput sequencing of the V3 to V4 variable regions of the 16S ribosomal RNA gene.

**Results**. Almost all phyla (91%) differed significantly (FDR < 0.05) in relative abundance according to treatment and there were distinct differences seen in community structure between treatment groups at this level. At family level, forty taxa showed significantly different relative abundance (FDR < 0.05). Bacteroidetes (*Bacteroidaceae*) and Firmicutes populations (*Lactobacillaceae*, *Clostridiaceae* and *Peptostreptococcaceae*) increased in relative abundance in the AA almond supplemented group. Supplementation with milk resulted in increased abundance of Actinobacteria (*Coriobacteriaceae* and *Bifidobacteriaceae*) compared with other groups. Soy supplementation increased abundance of some Firmicutes (*Lactobacilliaceae*) but not Actinobacteria, as previously reported by others.

**Conclusion**. Supplementation with milk or plant-based drinks has broad impacts on the intestinal microbiome of young rats. Changes induced by cow milk were generally

in line with previous reports showing increased relative abundance of *Bifidobacteriacea*, whilst soy and almond beverage did not. Changes induced by soy and almond drink supplementation were in taxa commonly associated with carbohydrate utilisation. This research provides new insight into effects on the microbiome of three commercially available products marketed for similar uses.

# INTRODUCTION

Core microbiomes have been identified in several species including rat, mouse and human (*Cullen et al., 2020*). The microbiome is influenced by diet and environment (*Rothschild et al., 2018*) and is integral for shaping of the host immune system (*Hooper, Littman & Macpherson, 2012*) and for gut homeostasis (*Gill et al., 2006*; *McFall-Ngai et al., 2013*). Dysregulation can lead to gastrointestinal diseases (*Ni et al., 2017*; *Rigottier-Gois, 2013*), neurologic disorders (*Suganya & Koo, 2020*), diabetes, obesity (*Dabke, Hendrick & Devkota, 2019*), rheumatoid arthritis (*Yoo et al., 2020*) and dysregulation of bone homeostasis (*Ibanez et al., 2019*; *Sjogren et al., 2012*). Dairy milk is generally considered an excellent source of nutrition (*Martínez-Padilla et al., 2020*). It has good digestibility and contains many nutrients that benefit the microbiome including protein (*Bai et al., 2016*; *Masarwi et al., 2018*), carbohydrates (*Estorninos et al., 2021*; *Kirmiz et al., 2018*) and fats (*Huang et al., 2013*). Fermented products from mammalian sources such as cheese and yoghurt are similarly beneficial (*Burton et al., 2017*; *Rettedal et al., 2019*). Beyond simple nutrition, the importance of milk for development of the gut microbiota during infancy, is well described (reviewed by *Laursen, 2021*).

Plant-based beverages continue to grow in popularity as alternatives to mammalian milk products. The nutritive value and digestibility of protein within plant-based products can vary significantly depending on the plant source (*Day, Cakebread & Loveday, 2022*; *Martínez-Padilla et al., 2020*) and brand (*Vanga & Raghavan, 2018*). That said, non-fermented and fermented soy beverage, like mammalian milk, is reported to have a comparable beneficial influence on the human gut microbiome (*Fujisawa et al., 2017*; *Inoguchi et al., 2012*). The newer non-dairy beverages, formulated from nuts (almond, hazelnut) grains (oat, rice) or legumes (pea) have yet to be studied. Almond milk is a mixture of fragmented almonds in water (~2%) which has been further processed to improve the suspension and microbial stabilities of the commercial product (*Vanga & Raghavan, 2018*). Almonds, as a whole or ground nut, have potential prebiotic properties with some reports showing an increase in *Bifidobacteria* (*Liu et al., 2014*; *Mandalari et al., 2008*) and another showing an increase in *Lachnospiraceae* populations (*Holscher et al., 2018*). Effects are influenced by processing and amount ingested (*Holscher et al., 2018*). To our knowledge, the impact of almond beverages on the microbiome has not been previously reported.

The aim of this research was to compare the impact of bovine milk, soy or almond beverages on the caecal bacterial microbiota in growing rats. We used two rodent feeds as controls; one contained intact casein protein and the other amino acids equivalent to casein. These groups were given water as their sole liquid source. The Casein water group (Casein water) represented healthy growing rats fed a replete diet representing the baseline control. The AA water group (AA water) represented an incomplete diet, also with water as their sole liquid. The treatment groups were fed the AA diet but were supplemented with bovine milk (AA milk), soy beverage (AA soy) or almond beverage (AA almond) as their sole source of liquid. All beverages were ultra-heat treated (UHT). In an earlier report from the same study (*Cakebread et al., 2019*) we demonstrated impact of supplementation on bone mineralisation, where supplementing the diet with bovine milk and soy beverage had favourable bone health outcomes. Conversely, almond beverage was not an effective supplement for bone mineralisation and bone strength outcomes, despite equal calcium intake to the Casein water group (*Cakebread et al., 2019*).

The gut microbiota is reported to be a major regulator of bone mass in mice (*Sjogren et al., 2012*) and we postulated that the soy and milk supplemented groups may have a more similar microbiota compared with the almond supplemented group.

Published studies show milk and soy supplements increase relative abundance of *Bifidobacteria* (*Favier et al., 2002*; *Fujisawa et al., 2017*; *Piacentini et al., 2010*) but the impact of commercial almond milk on the microbiome is unknown. This work compares the impacts of milk, soy and almond beverage supplementation on the caecal microbiome, as representative of the colonic microbiota (*Li et al., 2017*).

## METHODS

### Animal ethics

All animal experiments were performed in accordance with the guidelines of the New Zealand National Animal Ethics Advisory committee for the use of animals in research, testing, and teaching. All animal manipulations were approved by the Ruakura Animal Ethics Committee (AEC#14346), established under the Animal Protection (code of ethical conduct) Regulations Act, 1987 (New Zealand) as previously described (*Cakebread et al., 2019*). Animals were sourced from the Ruakura Small Animal Colony (Hamilton, NZ). All small animals bred for research are counted and contribute to AgResearch's overall animal use statistics, which are publicly available. Animal experiments can be carried out with either male or female animals (not both), because different genders may respond differently to treatments. For this study a total of 50 healthy, male Sprague-Dawley rats were used. At weaning, animals were randomly assigned into groups (3–4 per cage, and to give similar average group weights) and housed in specific-pathogen-free conditions in a temperature-controlled room with a 12 h light-dark cycle.

### Rodent diets and liquids

Standard AIN-93G rodent food containing casein protein (casein; 20%) and a modified version of AIN-93G food containing amino acids (AA) equivalent to the casein diet were purchased from Research Diets Inc., (New Jersey, USA) (*Reeves, Nielsen & Fahey, 1993*).

Diets were analysed by commercial testing (as described previously *Cakebread et al., 2019*) to confirm similarity of protein nitrogen, fat and mineral content (Table S1).

A range of UHT, unsweetened, unflavoured plant-based beverages were purchased from a local supermarket. Bovine UHT milk was used as the reference point to match the most comparable soy and almond plant-based drinks based on protein content and total calories for the feeding trial. Composition (protein, fat, total solids, energy, minerals) of the liquid supplements chosen for use in the trial were confirmed by commercial testing as described previously (*Cakebread et al., 2019*) (Table S2). Milk and beverages from the same batches were used during the trial. Beverage cartons were chilled to 4 °C before opening and open cartons stored at the same temperature (4 °C). Cartons were mixed well prior to aliquoting liquids into feeding bottles.

### Animal husbandry

Weanling male rats (3 weeks of age) were randomly assigned to five groups ($n = 10$/group) with similar weight distribution. The sample size of 10 per group was based on a nutritional experiment comparing goat and bovine milk, where the standard deviation of the main variable (bone mineral content) was 0.025. Power was at 80% and significance level (5%) to detect a difference of 0.033 (*Hodgkinson et al., 2018*). Two control groups were fed *ad libitum* either standard AIN-93G and water (Casein water) or modified AIN 93-G AA food and water (AA water). Three experimental groups were fed *ad libitum* AIN-93G AA food and either UHT bovine milk (AA milk), UHT soy beverage (AA soy) or UHT almond beverage (AA almond), as their sole liquid. Liquids were replaced daily, and food intake measured and replenished as required. Food and liquid intakes were recorded and used to calculate kcal intake over the duration of the trial (Table S3). Animals remained on their allocated food and liquid diets for four weeks. Rats were weighed twice per week to monitor health. Conditions for exclusion were established *a prioi* and included observation of general wellbeing (using the general health score for rodents) and <10% bodyweight loss over two consecutive weighings.  No unexpected events occurred, and all animals were included in the analysis.

### Sample collection

Caecum samples were collected from rats following euthanasia ($CO_2$ asphyxiation and cervical dislocation) at the age of seven weeks. Caecum contents were removed, and immediately snap frozen and then stored at −80 °C until further processing.

### DNA extraction and sequencing

Bacterial DNA was extracted from each caecum using NucleoSpin® Soil kits (Machery-Nagel, ThermoFisher Scientific, Auckland, NZ) according to the manufacturer's instructions. Microbiota profiling was assessed using barcode paired end 2 × 250 bp sequencing of bacterial 16S rRNA gene PCR products (Illumina MiSeq sequencing, Massey Genome Service, Massey University, Palmerston North, NZ) targeting the V3 and V4 hyper-variable region (primers listed in Fig. S1). Prior to sequencing, samples were QC checked using Invitrogen Quant-iT dsDNA HS Assay and quantified using a Qubit® Fluorometer.

Sequences were processed following a modified form of the pipeline described in (*Camarinha-Silva et al., 2012*). The reads produced by the sequencing instrument were paired using the program FLASH2 (*Magoc & Salzberg, 2011*). Paired reads were then quality trimmed using Trimmomatic (*Bolger, Lohse & Usadel, 2014*). The trimmed reads were reformatted as fasta, and the read headers were modified to include the sample name. All reads were compiled in a single file, and the Mothur (*Schloss et al., 2009*)program suite was used to remove reads with homopolymers longer than 10 nt, and to collapse the reads into unique representatives. The collapsed reads were clustered with the program Swarm (*Mahe et al., 2014*). The clustered reads were filtered based on their abundance, keeping representatives that were (a) present in one sample with a relative abundance >0.1%, (b) present in >2% of the samples with a relative abundance >0.01% or (c) present in 5% of the samples at any abundance level. The selected representatives were annotated using the QIIME program with the Silva database for bacteria, and RIM-DB for archaea. The annotated tables were then used for downstream statistical analysis. A Ruby program implementing the abundance filter is available in the AgResearch Gitea website (https://github.com/ruy-jauregui/microbiomics). Raw reads have been deposited at the ncbi under the bioproject number PRNJA782341. Samples are listed at https://www.ncbi.nlm.nih.gov/sra?linkname=bioproject_sra_all&from_uid=782341.

## Statistical analysis

Body weights were compared by treatment using ANOVA in Genstat (Genstat for Windows 17th edition; VSN International) and with trial day 0 weight as a covariate. Means were compared using Fisher's unprotected least significant difference test and $P$ values < 0.05 were considered significant.

Statistical analyses of microbiota were performed using R 4.0.3 (*R Core Team, 2020*). Cage was included as a random effect. Differences between relative abundances of individual taxa among the different treatments were assessed for significance using permutation analysis of variance as implemented in the lmPerm package in R (*Wheeler & Torchiano, 2016*). Taxa with an FDR <0.05 were considered significantly different.

# RESULTS

## Treatment groups

Differences between group kcal intakes were highly correlated with group weight ($R^2$ range 0.87–0.99), as previously reported (*Cakebread et al., 2019*). Briefly, at the start of the trial, group average weights of weanling rats were not significantly different (47.5 g: range 34.2–61.6 g). At the end of the trial group average weights of animals in Casein water and AA milk groups were similar (291 g and 297 g, respectively). Both groups were significantly heavier than animals in AA water and AA almond groups (251.2 g and 215.2 g, respectively). Animals in the AA almond group were significantly lighter than all other groups. In contrast, the average weight of AA soy group was significantly higher than all other groups (326 g). Weights, intake, and calculated estimates of macronutrients are listed in Table S3.
## Comparison of overall community structure: phyla

Illumina sequencing returned a total of 10,329,465 sequences for the caecal microbiome of the fifty rats across the five supplement groups, with a minimum library size of 78,645, mean of 206,589, and maximum of 427,702 reads.

The main phyla observed in the rat caecal microbiota were Firmicutes, Bacteroidetes, Actinobacteria, and Proteobacteria, representing over 97% of the total population, which is in line with the reported literature (*Li et al., 2017*).

Almost all phyla (91% of sequence reads) differed significantly (FDR <0.05) in relative abundance according to treatment and there were distinct differences seen in community structure between treatment groups at this level (Table S4). Of note was the striking increase in relative abundance of Actinobacteria in the AA milk group (21%) compared to all other groups (range 1.3–2.2%). Firmicutes and Bacteroidetes were the most prevalent phyla across all groups (relative abundance range 52%–67% and 20–28%, respectively). The AA milk group had significantly lower relative abundance of Firmicutes (52% compared to the other groups (range 61–67%)) and Bacteroidetes (20%, compared to all other groups (range 25–28%, Table S4)). Proteobacteria were more abundant in the AA soy group (9%) compared to the other groups (range 5–6%). An increased abundance of Acidobacteria was observed for the AA almond group (0.69%) compared to AA soy (0.11%), AA water (0.24%) and Casein water (0.08%), but it was present in low amounts and not detected in the AA milk group.

## Comparison of overall community structure: family

At the family level, forty taxa showed significantly different relative abundance (FDR < 0.05, Fig. 1). The most different, based on highest proportions, are shown in Table S5.

Of the phylum Actinobacteria, *Coriobacteriaceae* (0.24% Casein water, 1.6% AA water, 14.9% AA milk, 1.1% AA soy and 1.7% AA almond) and *Bifidobacteriaceae* (0.4% Casein water, 0.1% AA water, 6.0% AA milk, 0.05% AA soy and 0.4% AA almond) were notably greater in relative abundance for the milk supplemented group (Fig. 2, Table S5).

Bacteriodales *Porphyromonadaceae* (Fig. 3) and *Bacteriodales S24-7 group* (Fig. 1, Table S5) had a significantly lower relative abundance in the AA milk group (1.29% and 3.47%, respectively) compared to all other groups (range 2.75–3.42% *Porphyromonadaceae* and 6.26–9.79% *Bacteriodales S24-7 group*, Fig. 3). Conversely, there was an increase in abundance of *Bacteriodaceae*, in AA milk (11.86%) and AA almond (11.97%) compared to AA soy (6.01%; Fig. 3).

Treatments impacted many taxonomic families in phylum Firmicutes (Fig. 4). For example, one taxon, *Clostridiaceae group 1* had higher relative abundance in the AA water group (7.3%) compared the Casein water group (3.7%) suggesting an effect of the base diet. A reduction in abundance was observed in the AA almond group (5.16%) compared to AA water (7.3%) suggesting an effect from the supplement. A further reduction in abundance of *Clostridiaceae group 1* abundance were observed in rats fed AA milk and AA soy diets (1.63% and 0.15%, respectively). The AA milk and AA soy groups also had a lower relative abundance of *Peptostreptococcaceae* (12.37% and 12.36%, respectively) compared to AA almond (17.3%), and also AA water and Casein water groups (19.78%

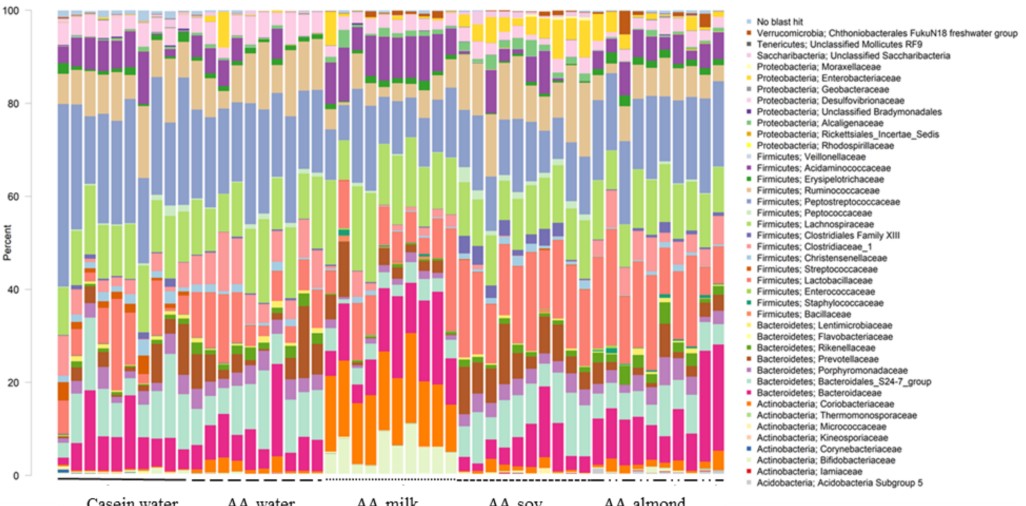

**Figure 1** **Changes to bacterial family communities.** Changes to bacterial communities with different liquid supplements. The relative abundance (family) present in caecal contents collected from individual rats ($n = 50$) are shown. Two control groups received water with either casein diet (Casein water) or amino acid diet (AA water). Treatment groups received AA diet with either bovine milk (AA milk), soy beverage (AA soy), or almond beverage (AA almond). Diets and treatments were supplied over 4 weeks from weaning to 7 weeks old. The colours represent different families as indicated in the figure legend.

and 24.21%, respectively). *Lactobacillaceae,* had higher abundance in the AA soy (16.8%) and AA almond (15.63%) supplemented groups compared to the other groups (Casein water 7.19%, AA water 7.89% and AA milk 9.06%). *Lachnospiraceae* had higher abundance in the AA milk (14.4%) compared to all other groups supplements (Casein water 12%, AA water 12.4%, AA soy 10.9% and AA almond 11.0%).

Proteobacteria family member *Enterobacteriaceae* (Fig. 5) were significantly higher in relative abundance for the AA soy supplemented group (4.6%) compared to all other groups (Casein water 0.1%, AA water 0.98%, AA milk 1.0%, and AA almond 2.0%). *Desulphovibrionaceae* had lower abundance in groups given the supplements (AA milk 2.75%, AA soy 2.62% and AA almond 2.44%) compared to controls (Casein water 3.96% and AA water 4.51%) suggesting a supplement treatment effect.

## DISCUSSION

This study, the first to our knowledge, compares effects of commercial soy and almond plant-based beverages and bovine milk supplementation on the caecal microbiota in a growing rat model. Our main finding was the significant increase in Actinobacteria (*Bifidobacteriaceae* and *Coriobacteriaceae*) and Firmicutes (*Lachnospiraceae)* in the UHT milk fed group relative to the other groups. This was not observed for soy beverages as previously reported in human studies (*Fujisawa et al., 2017*; *Piacentini et al., 2010*). No previous studies were found for almond beverage.

Human studies of whole or processed almonds have shown increases in relative abundance of Actinobacteria (*e.g.*, *Bifidobacterium spp.* and *Lactobaccillus spp)* and

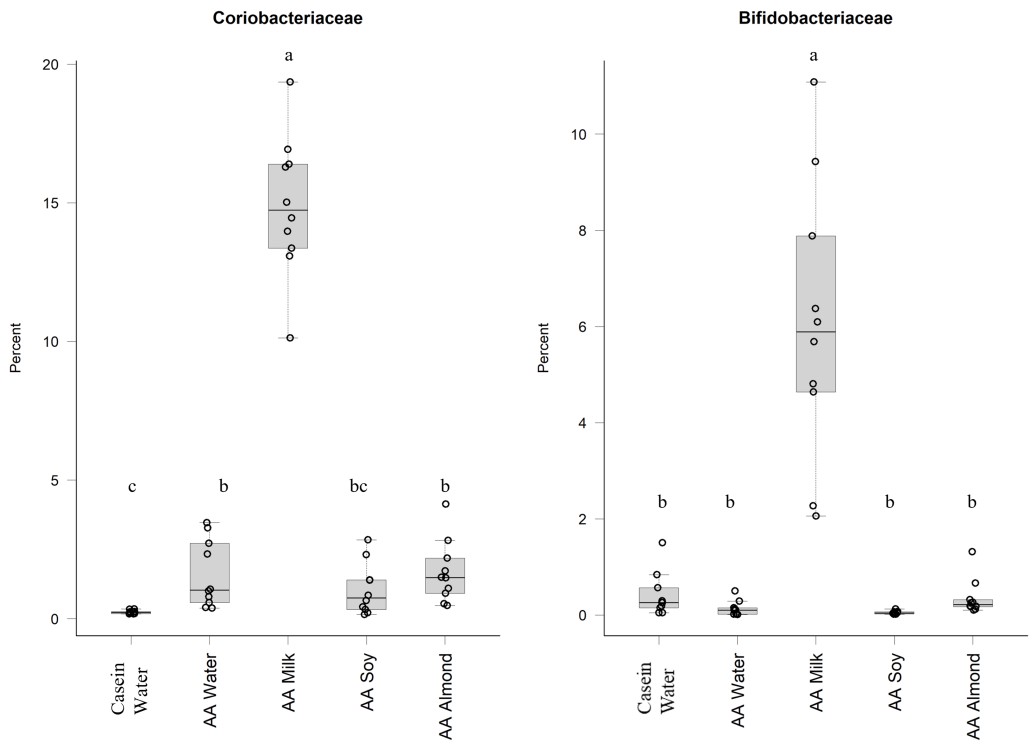

**Figure 2** **Relative abundance (%) of Actinobacteria for the five groups.** Two control groups received water with either casein diet (Casein water) or amino acid diet (AA water). Treatment groups received AA diet with either bovine milk (AA milk), soy beverage (AA soy), or almond beverage (AA almond). Differences between individual taxa among the different treatments were assessed for significance using permutation analysis of variance. Dissimilar letters denote significant differences ($P < 0.05$). Midline shows median, the upper and lower limits of the box showing the third and first quartile (*i.e.,* 75th and 25th percentile), respectively and the whiskers represent 1.5 times the interquartile range. Open circles represent outliers (*i.e.,* $>1.5 \times$ IQR). $n = 10$ per group.

Firmicutes (*e.g.,* *Lachnospira* (genus) (*Holscher et al., 2018*; *Liu et al., 2014*). It has been suggested that dietary fibre and polyphenols could be associated with microbiota effects (*Liu et al., 2014*) but levels of these were unknown for this study. The amount of almond found in the almond beverage is much less compared to that given in the human studies (*Holscher et al., 2018*; *Liu et al., 2014*; *Vanga & Raghavan, 2018*) and suggests this would reduce the likelihood of seeing an effect.

Bacteroides and Firmicutes have a well-documented ability to utilise carbohydrates (*El Kaoutari et al., 2013*). The estimated carbohydrate intake for each group is shown in Table S3 and indicates a lower intake for the AA almond group over the course of the study. Alterations in the type and quantity of polysaccharides consumed can result in changes in the microbiota community composition and function (*Sonnenburg et al., 2010*). We saw an influence of almond supplementation on *Bacteroidaceae* (Phylum Bacteroidetes), despite lower carbohydrate intake. Bacteroidetes members share a common ancestor (*Pace, 1997*) and have gene-encoded carbohydrate active enzymes that can switch readily between different energy sources in the gut, depending on availability (*Flint et al., 2012*). Aside from

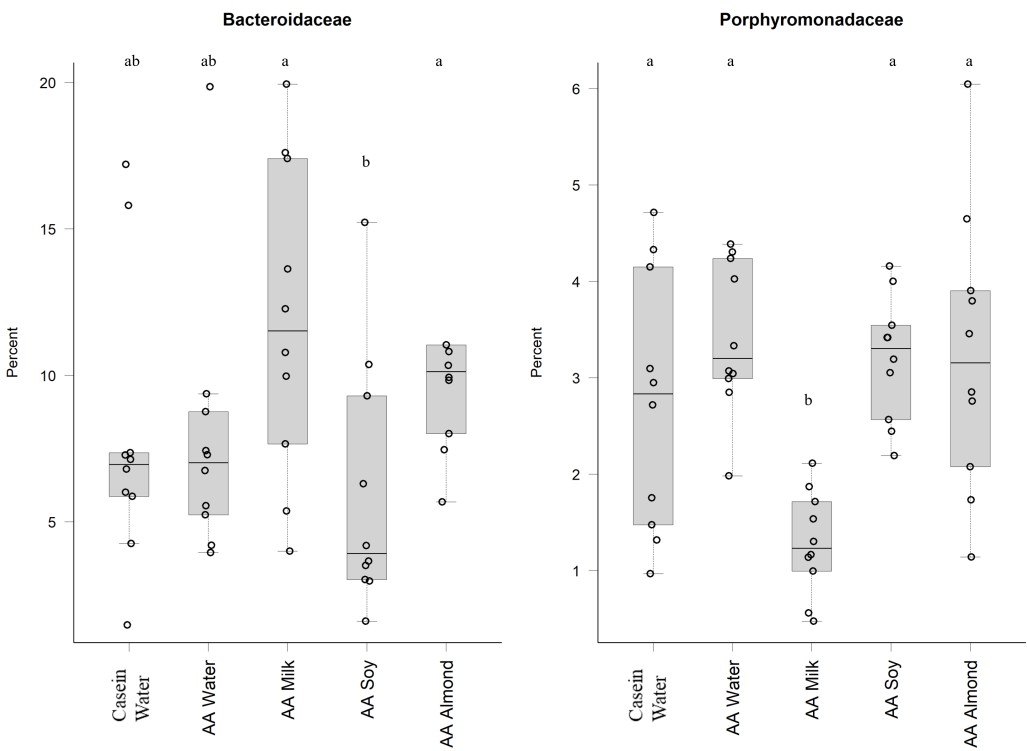

**Figure 3** **Relative abundance (%) of Bacteriodetes in the five groups.** Two control groups received water with either casein diet (Casein water) or amino acid diet (AA water). Treatment groups received AA diet with either bovine milk (AA milk), soy beverage (AA soy), or almond beverage (AA almond). Differences between individual taxa among the different treatments were assessed for significance using permutation analysis of variance. Dissimilar letters denote significant differences ($P < 0.05$). Midline shows median, the upper and lower limits of the box showing the third and first quartile (*i.e.,* 75th and 25th percentile), respectively and the whiskers represent 1.5 times the interquartile range. Open circles represent outliers (*i.e.,* $>1.5 \times$ IQR) $n = 10$ per group.

amount ingested, the type of carbohydrate may have influenced the bacterial populations, since the predominant carbohydrate in soy and almond supplements are non-starch polysaccharides (NSP) and sugars (*Choct et al., 2010*)whereas for milk it is lactose (*Roy et al., 2020*).

Metabolites produced by microbial populations and cross-feeding has the potential to affect bacterial community assembly and stability characteristics (*Rios-Covian et al., 2013*). For example, members of the genus *Bifidobacterium* have a superior ability to sequester oligosaccharides found in milk and soy (*Ma et al., 2017*; *Sela & Mills, 2010*) and can successfully compete with members of the genus Bacteroides. Furthermore, we observed variations in Firmicutes populations for the almond supplemented group showing increased abundance of *Lactobacillaceae*, *Clostridiaceae_1* and *Peptostreptococcaceae*. Firmicutes are known to play a key role in nutrition and metabolism of the host through short chain fatty acid (SCFA) synthesis (*Stojanov, Berlec & Štrukelj, 2020*) that may contribute to the pathophysiology of obesity (*Riva et al., 2017*) although this is still under debate (*Magne et al., 2020*). However, in this study AA almond and AA water groups were significantly

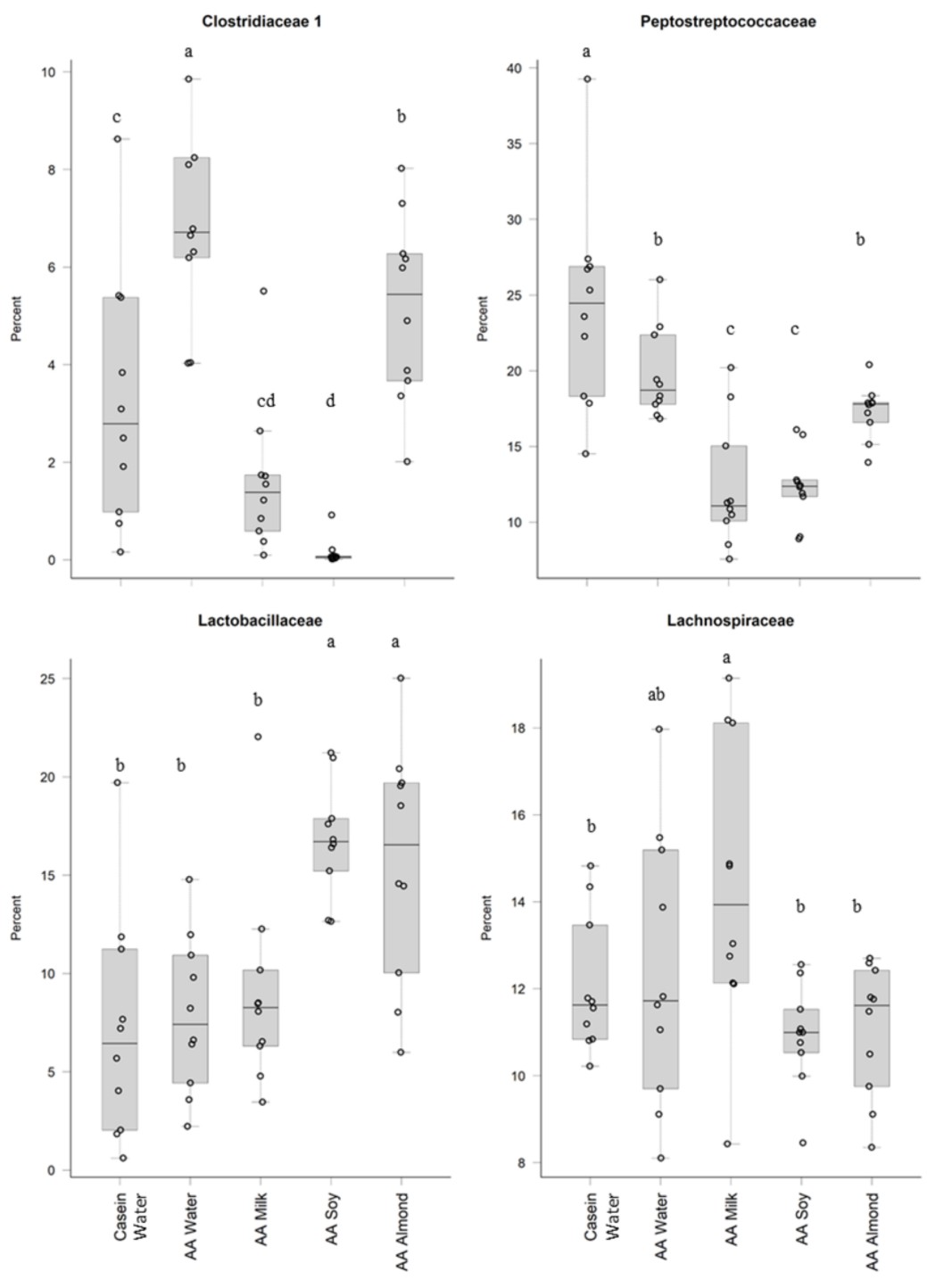

**Figure 4  Relative abundance (%) of Firmicutes in the five groups.** Two control groups received water with either casein diet (Casein water) or amino acid diet (AA water). Treatment groups received AA diet with either bovine milk (AA milk), soy beverage (AA soy), or almond beverage (AA almond). Differences between individual taxa among the different treatments were assessed for significance using permutation analysis of variance. Dissimilar letters denote significant differences ($P < 0.05$). Midline shows median, the upper and lower limits of the box showing the third and first quartile (*i.e.,* 75th and 25th percentile), respectively and the whiskers represent 1.5 times the interquartile range. Open circles represent outliers (*i.e.,* $>1.5 \times$ IQR) $n = 10$ per group.

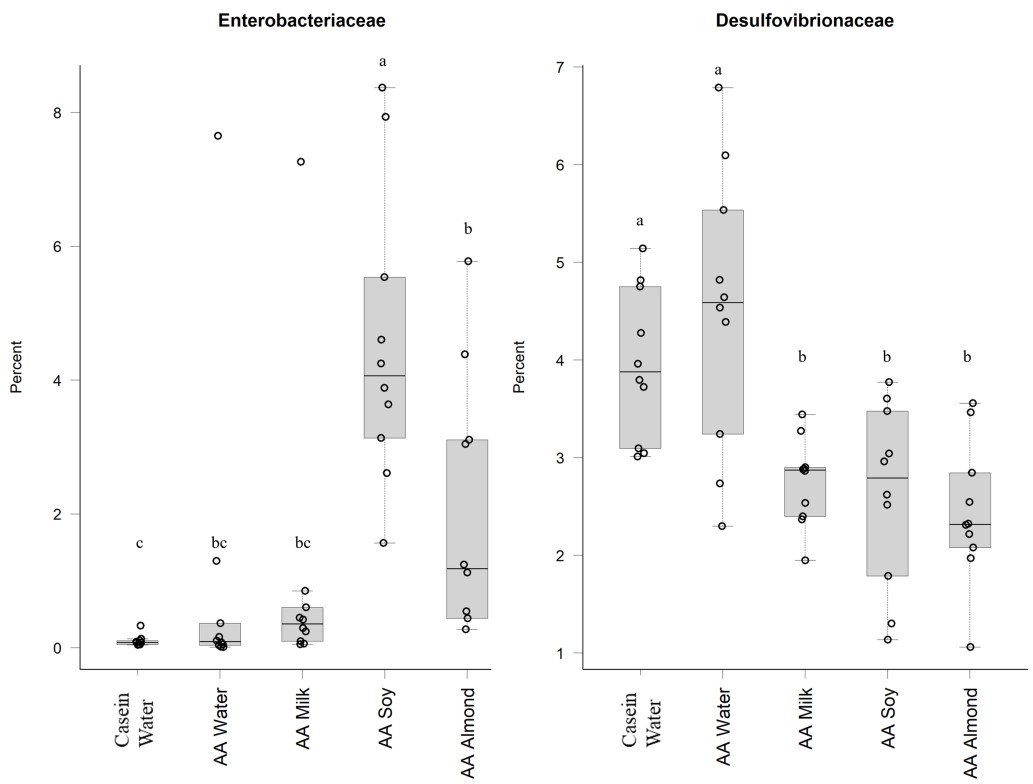

**Figure 5  Relative abundance (%) of Proteobacteria in the five groups.** Two control groups received water with either casein diet (Casein water) or amino acid diet (AA water). Treatment groups received AA diet with either bovine milk (AA milk), soy beverage (AA soy), or almond beverage (AA almond). Differences between individual taxa among the different treatments were assessed for significance using permutation analysis of variance. Dissimilar letters denote significant differences ($P < 0.05$). Midline shows median, the upper and lower limits of the box showing the third and first quartile (*i.e.,* 75th and 25th percentile), respectively and the whiskers represent 1.5 times the interquartile range. Open circles represent outliers (*i.e.,* $>1.5 \times$ IQR). $n = 10$ per group.

lighter than other groups reflecting the reduced calorie intake (Table S3) (*Cakebread et al., 2019*).

The relative abundance of *Bifidobacteriaceae* and *Coriobacteriaceae* was over fifteen times and eight-times higher, respectively, in the AA milk group compared to other groups. An abundance of *Bifidobacterium* has been demonstrated in breast-fed human infants, which is attributed to the presence of glycans (including carbohydrates, sugars, monosaccharides, oligosaccharides and polysaccharides) which are readily utilised by the bacteria. The lactose found in milk can be utilised by these bacteria (*Gonzalez-Rodriguez et al., 2013*) and could be a major contributing factor in the observed increased abundance. Furthermore, prebiotics derived from lactose (*e.g.,* galacto-oligosaccharide and fructo-oligosaccharide) have been shown to have positive effects on calcium absorption although the mechanisms are not well defined (reviewed by *Macfarlane, Macfarlane & Cummings, 2006*). Other glycans in milk occur as free oligosaccharides (*Marcobal & Sonnenburg, 2012*; *Robinson, 2019*)or as glycoconjugates (*Kirmiz et al., 2018*) that can also act as prebiotics to stimulate

the growth of anaerobes of the genera *Lactobacillus* and *Bifidobacterium* (*Albrecht et al., 2014*; *Underwood et al., 2015*). The benefits of adding bovine milk oligosaccharides to infant formula has been demonstrated (*Estorninos et al., 2021*).

Oligosaccharides, specifically raffinose and stachyose, are also found in soybeans, and can be used by *Lactobacilliaceae* as an energy source (*Chen et al., 2021*). This could explain the increased abundance shown in the current study in the AA soy group. Another study supports this, and showed oral doses of a soy oligosaccharide extract increased *Bifidobacterium* abundance in adult men (*Hayakawa et al., 2009*). However, in our study we saw no such increase in the soy treatment group (Table S5). The soy supplemented group had higher relative abundance of the Proteobacteria (*Enterobacteriaceae)* which has been observed previously (*An et al., 2014*), and has been associated with obesity and dysbiosis in human populations (*Mendez-Salazar et al., 2018*; *Shin, Whon & Bae, 2015*) and we noted the soy group had the highest weights (Table S3) (*Cakebread et al., 2019*). The increased abundance of Proteobacteria in the soy group in our study was driven mostly by an increase in abundance of *Enterobacteriaceae,* whilst *Desulfovibrionaceae* were reduced in all the supplemented groups (milk, soy, and almond) compared to the Casein water and AA water groups.

Milk supplementation resulted in increased Actinobacteria (*Bifidobacteriaceae* and *Coriobacteriaceae)*, with lower abundance of Bacteroidetes and Firmicutes, previously associated with a balanced gut homeostasis (reviewed in *Binda et al., 2018*). The almond beverage had most effect on Bacteroidetes and Firmicutes populations, whilst soy affected Proteobacteria families.

Many soy and milk studies have been performed (mostly using powders or concentrates), but none have not compared liquid UHT milk, soy and almond beverages. For example, *An et al. (2014)* fed milk casein and soy protein (both at 20%) as a solid feed in a rat model, whilst Butteiger et al. delivered milk protein isolate, soy protein isolate and soy concentrate (22%) (*Butteiger et al., 2016*) in a solid diet and in a golden Syrian Hamster model. Published studies investigating effects of almonds have inconsistent experimental design with variations in age of cohort, dose and formulation (*Holscher et al., 2018*; *Liu et al., 2014*). Our study used UHT liquid products as an *ad libitum* supplement in a growing rat model, which is novel, and makes direct comparisons between earlier studies challenging.

We note that the bone density results from our study suggests milk and soy supplementation are equally beneficial for (bone) health (*Cakebread et al., 2019*) whereas the microbiota data suggests the three supplements develop distinct microbiota profiles. It is challenging to relate these findings to a 'health conclusion' since we describe only the bacterial community profile using amplicons of the 16S gene. A more comprehensive analysis including metabolic profiling and functional analysis is possible, but this is dependent on the availability of whole genome sequencing data which remains for the moment a future prospect in this project.

## CONCLUSION

The composition of the gut microbiota is determined by dietary input and by the metabolic outputs of the resident microbiota. This is the first study looking at three commercially

available beverages, marketed as comparable. We demonstrate clear differences in microbiota populations for these three products and provide new insight into effects of almond beverage which has not previously been reported. We are only beginning to look beyond nutrition to investigate the magnitude of impact the gut microbiome has on its host. To design efficacious plant-based foods with consistent biological effects, and which are comparable or better than the originals, further studies are needed. These studies should involve a 'system approach' to understand and model the complexity of the biological system that is human nutrition and health.

### Funding
This work was supported by the Ministry of Business Innovation and Employment, New Zealand, through AgResearch's Strategic Science Investment Fund. The funders had no role in study design, data collection and analysis, decision to publish, or preparation of the manuscript.

### Grant Disclosures
The following grant information was disclosed by the authors:
Ministry of Business Innovation and Employment, New Zealand, through AgResearch's Strategic Science Investment Fund.

### Competing Interests
All the authors declare that they have no competing interests. At the time of the study all authors worked for AgResearch ltd.

### Author Contributions
- Julie Cakebread conceived and designed the experiments, performed the experiments, analyzed the data, prepared figures and/or tables, authored or reviewed drafts of the paper, and approved the final draft.
- Olivia A.M. Wallace performed the experiments, authored or reviewed drafts of the paper, and approved the final draft.
- Harold Henderson analyzed the data, authored or reviewed drafts of the paper, and approved the final draft.
- Ruy Jauregui analyzed the data, authored or reviewed drafts of the paper, and approved the final draft.
- Wayne Young analyzed the data, prepared figures and/or tables, authored or reviewed drafts of the paper, and approved the final draft.
- Alison Hodgkinson conceived and designed the experiments, authored or reviewed drafts of the paper, and approved the final draft.

### Animal Ethics
The following information was supplied relating to ethical approvals (*i.e.*, approving body and any reference numbers):

All animal experiments were performed in accordance with the guidelines of the New Zealand National Animal Ethics Advisory committee for the use of animals in research, testing, and teaching. All animal manipulations were approved by the Ruakura Animal Ethics Committee (AEC#14346).

## Data Availability

The sequences are available at NCBI: SAMN23375632–SAMN23375681, PRJNA782341.

## Supplemental Information

Supplemental information for this article can be found online at http://dx.doi.org/10.7717/peerj.13415#supplemental-information.

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
