# Peer review of "The impacts of bovine milk, soy beverage, or almond beverage on the growing rat microbiome"

_PeerJ, doi:10.7717/peerj.13415_

## Round 0.1 · original submission · Major Revisions

Reviewers have commented against the acceptance of manuscript in its current form. Manuscript suffers from serious concerns regarding the implemented protocol as well as presentation and availability of the data, which requires substantial and thorough revision in order to appreciate the quality. Please revise in light of reviewers comments and resubmit accordingly.

Reviewer 1 ·

Basic reporting

1. I could not access https://gitea.agresearch.co.nz/JAUREGUIR/Microbiomics

2. I did not see a link to a public repository in which the authors deposited their data.

Experimental design

1. While the manuscript is of potential interest, this choice of dietary treatment greatly distracts from its significance. The authors should have chosen milk, milk formula and soy formula.

2. Both males and females mut be studied.

3. The authors need to state how random assignment to groups was achieved.

4. The authors need to test for heterogeneity of variances prior to conducting statistical analysis.

5. Why use rats as a model? This study should have been conducted in humans.

6. The observation that diet alters the gut microbiome is not novel.

7. I might have overlooked it, but how did the authors account for cage effects.

8. The study is descriptive and does not offer insights into functional consequences.

Validity of the findings

No comment.

Reviewer 2 ·

Basic reporting

Please see "additional comments" below to see all my comments and suggestions.

Experimental design

Please see "additional comments" below to see all my comments and suggestions.

Validity of the findings

Please see "additional comments" below to see all my comments and suggestions.

Additional comments

Cakebread et al. investigated the impact of bovine milk, soy beverage, or almond beverage on the growing rat microbiome. Two control groups were used (AIN-93G food, 20% casein protein; AIN-93G amino acids food). Animals on AIN-93G AA food were offered three diets (ultra-heat treatment [UHT] milk, and UHT soy and almond beverages) were offered for a period of 4 weeks.

General comments
Please make sure that you allow an easy identification of all 5 groups. For instance, it is difficult for me to guess which group in Figure 1 match the description of groups in the abstract and in the methods. I assume that AA water is the group with AIN-93G AA food that was not supplemented with either milk, soy or almond?
I did not see anything about cecum weight or cecum weight contents, could you please explain why you did not study these variables? See this for a reference about these variables: https://pubmed.ncbi.nlm.nih.gov/28815303/

Line 59. It is Bifidobacteriaceae, as the authors rightfully placed in other areas.
Lines 145-147. Why exactly did you choose to offer the diets to the animals on AA food and not to the casein control group?
Line 155. Is there any reason why you choose caecum samples only?
Line 159. Caecum contents? Or did you use the whole tissue?
Line 177. QIIME, with capital letters.
Line 183. The incorporation of covariates is somehow complex (https://genstat.kb.vsni.co.uk/knowledge-base/tancova/; https://www.stat.purdue.edu/~bacraig/notes514/topic10a.pdf), in part for the possible differences in slopes among treatments. However, I do not see anything in the results or discussion about differences among slopes. See the barley paper mentioned above. Also, is there any reason why you decided not to use baseline microbiota composition as a covariate in the microbiota analysis?
Lines 198-199. Is the casein water group the same as the UHT milk treatment group? How do you explain this finding? In other words, how the AA water and AA almond groups ended up related to lower body weights?

Lines 257-258. Please re-structure the discussion, these two lines look odd by themselves. Also, the next paragraph of 5 lines could also be incorporated more nicely with other parts of the discussion.
Line 259. Please make sure you only use the correct names associated with the right taxonomic level. Bifidobacteria is incorrect.
Line 260. Why do you talk about UHT milk fed groups, in plural? I had the impression that only one treatment group received AIN-93G AA food plus UHT milk.
Line 324. Effects? Bacteroidetes? You are talking about taxa at the phylum level, am I correct?
Line 325. Does not this paragraph contradict the previous statement in lines 257-258? Please explain.
Line 337. What do you mean by “linear labelling”?
It would be nice to incorporate a paragraph describing the limitations of this study, for example the use of only one technique to analyze the microbiota, or the fact that the authors decided not looking at the prediction of metagenome functional content, e.g. using PICRUSt.

References
Please be consistent with the references, in some the first letter of each word is capitalized.

---

## Round 0.2 · accepted · Accept

The manuscript is significantly improved by the authors and now can be accepted in its current form.